## [Transparent Peer Review File · Nature Communications]

NF- κ B activation in astrocytes critically affects wound healing after traumatic brain injury

Corresponding Author: Dr Bernd Baumann

Version 0:

Reviewer comments:

Reviewer #1

(Remarks to the Author)

In this manuscript entitled, NF κ B activation in astrocytes critically affects wound regeneration after TBI, the authors use a weight-drop model of closed-skull brain injury (in mice) to explore gain of function (GoF) and loss of function (LoF) of NF κ B signaling (IKK2) in astrocytes and focus on wound healing and inflammation. This closed skull injury model induces a significant wound in the cortex of mice. Here, the authors report that TBI induces a prominent NF κ B-related gene expression signature within the cortex after TBI. This corresponds with proliferation of astrocytes, cortical inflammation, and induction of a wound healing/glia scar response. Based on the initial analyses, the authors predict that astrocytic responses in NF κ B signaling are important in the inflammatory response to TBI. As such, they use astrocytic-IKK2 gain of function (GoF) and astrocytic-IKK2 loss of function transgenic lines. They report that TBI-WT, TBI-CA (gain of function), and TBI-DN (loss of function) all result in different transcriptomic signatures in astrocytes. In general, the astrocytic-IKK2 gain of function mice had worse cortical lesions and increased myeloid cell infiltration compared to the WT-TBI mice. Nonetheless, the astrocytic-IKK2 (GoF) had major effects on transcriptional profiles, independent of TBI. While, astrocyte-NF κ B GoF increased the size of the lesion initially (up to 15 dpi), the wound sizes were similar by 30 dpi. The dominant negative of IKK2 in astrocytes did not have much of an influence on the parameters tested. This may reduce the relevance of the data using the IKK2 overexpression in astrocytes. The overall conclusion is that over expression of IKK2 in astrocytes disrupts wound healing and this is associated with more inflammation and recruitment of myeloid cells.

1. One major concern is that in that the gain of function experiments using astrocytic-IKK2 show that there is a robust effect of genotype, independent of TBI. This effect is clear in the data from figures 4 and 5. In figure 5, the GoF in astrocytes induces a TBI-like response in the context of the transcriptional analyses. While the authors acknowledge this, it is still problematic for the study. One issue is that the gain of function of IKK2 (or overexpression) in astrocytes seems artificial (based on the results with the dominant negative version) and therefore, less biologically relevant. In addition, the study focuses only with the lesion area of the cortex after TBI. The gain of function problem in the astrocytes will be widespread throughout the brain. Furthermore, the potential confounds of this main effect of genotype is obscured throughout the paper. The authors switch from showing sham-controls to showing all TBIs groups with comparisons of the ipsilateral vs contralateral sides of the injured cortex (Figs.7-9). It is important to note that the authors show an interaction effect between astrocytic-IKK2 and TBI in the context of lesion size (up to 15 dpi), myeloid cell infiltration in the lesion, and with the transcriptional analyses. For the lesion, however, it is the same between TBI and TBI-astrocytic-IKK2 by 30 dpi. It is also unclear how the data in Fig.8 supports the author's claim on wound healing. These data are difficult to interpret and they lack sham controls. For these reasons, the primary conclusion of the paper are not convincingly supported by the data presented.

2. Another related concern is that astrocytic-IKK2 dominant negative (IKK2-DNGFAP) does not affect TBI induced CNS scar formation or myeloid cell infiltration. The transcriptional effects in the astrocytic-IKK2, seem different (from the PCA plot) but were not explored in the manuscript. Based on the data in Fig.1 and the assumptions of the introduction, the astrocytic-IKK2 dominant negative should reduce the cortical inflammation after TBI, but it does not. Because it does not seem to have much of a role normally, the overexpression of IKK2 in astrocytes has reduced biological significance.

3. There are also some concerns on novelty and relevance of the study. For example, it is well established that TBI causes inflammation and increases in NF κ B mediated pathways in the brain of rodents. Moreover, myriad neurotrauma (SCI and TBI) studies in rodents show that astrocytes are critical for glia scar formation, are essential for wound healing, and influence immune cell infiltration in the brain and spinal cord. Here, the main findings is that a gain of function/overexpression of NF κ B

signaling in astrocytes impairs the wound healing response at the lesion site.

4. Another concern is that the description for the generation of the dominant negative (LoF) and constitutively activated (GoF) cell specific mouse lines are not included within the manuscript. Also, the authors don't provide any validation that the NFkB pathway is overexpressed in astrocytes and that the NFkB pathway is non-functional in astrocytes. Validation of the models is an important element of rigor and reproducibility.

5. In the assessment BrdU+ IBA1 and GFAP at 3dpi, there are no sham groups included.

6. In Fig.2 the authors only show representative images at 3 dpi. Authors should show all time points, not just 3dpi, especially because it seems to peak at 7dpi.

7. Concern that in Fig.3 the text does not match the data/representative images. Authors state that controls increase from day 1 to day 3, but it seems to decrease in the image and not really change in the graph. As they say, the lesion decreases in controls from day 3 to day 7 – 15 but increases and then stays the same in the GoF. In Fig.3b, there is a key for all three groups but then they do not include the LoF group. It also looks like the representative image do not match the graph for 1dpi. It is also unclear why Fig.3c separates all the groups and what time point this graph represents.

8. Fig. 6. shows volcano plots for TBI-WT v sham and TBI-CA v sham but not direct comparison, even though that is the group of most interest.

9. Concern that the data in Fig.7 are disjointed. It starts with data from fig 6 and then moves to blots of ipsi/contra tissue and then ending with more astrocyte transcriptional data from before. It is unclear if control is contralateral tissue and CA is ipsi or if this is just a genotype comparison. Also unclear why in Fig.7d-f, it switches from including sham controls to contralateral controls.

10. Fig. 9 shows western blots of proteins they say are important for wound healing and inflammation. Nonetheless, no citations supporting this idea are provided. The author's use of the ratio of osteopontin (protective) to lipocalin-2 (inflammatory) is arbitrary.

Reviewer #2

(Remarks to the Author)

In the submitted manuscript, Hein et al use astrocyte-specific strategies to modulate NfKB activation through loss of function(IKK2-DN GFAP) or gain of function(IKK2-CA GFAP) mouse models and investigate the how this affects various outcome in a closed head injury model of TBI.

An important role for NfKB following traumatic injury is long established, though the attempt to look at this from an astrocyte-specific perspective is interesting and novel.

A large proportion of the data in the paper is from bulk transcriptomic analysis of either whole tissue or isolated cell populations. Much of this can be confusing as direct comparisons are often made between groups that are of secondary interest and then descriptive comparisons of these comparisons are made (see below).

The main finding of the paper is that IKK2-CA GFAP exacerbates many readouts after TBI including behaviour, tissue lesion size and, most interestingly, immune cell recruitment.

The title of the paper talks about 'wound regeneration' but the paper doesn't really get into what 'regeneration' means and it data to show actual wound regeneration is lacking. It would be good if the authors could clarify what they mean by 'wound healing' and 'wound closure' and what this means in the brain?

Also, a major theme of the paper is scar formation and tissue remodelling, and there is transcriptional and some protein analysis to support this, though a much more thorough analysis of the scar and tissue in tissue sections would be needed to support this (see below).

The major interesting finding in this paper is that astrocyte-specific activation of NfKB is sufficient to massively induce a peripheral infiltration of immune cells that are likely responsible of increase tissue damage and lack of resolution of inflammation. This has not been shown before and is of interest to the field.

Lastly, the authors largely focus on gain of function(IKK2-CAGFAP) rather than loss of function IKK2-DNGFAP as the latter had little to no effect on the model. How does this fit with the authors view that NfKB could be an effective therapeutic target?

(Note: On multiple occasions figures are not referred to in the text.)

I have multiple comments that would also need addressing, in figure order:

Figure 2

Figure 2e authors describe microglia maintaining NfκB expression but an alternative explanation could be that Iba1 positive recruited monocyte-derived macrophages are taking their place.

The authors need to do experiments to show definitively microglia are changing, or tone down microglial-specific conclusions

Figure 3

Fig. 3b measures 'Trauma dimension' and 3c is 'TBI area'. Fig 3c certainly is underpowered if the authors want to claim a no change.

Fig. 3c shows control vs IKK2-DNGFAP but this is not referred to in the text at all.

Fig. 3d How do the authors account for what appears to be other lesion sites across the brain when quantifying lesion volume? Ex vivo imaging clearly shows major cavities proximal and distal to the arrow to where they indicate lesion was measured.

A reduction in lesion volume at 30 days does not equate to tissue regeneration. Is a collapsed cavity or tissue swelling/reorganisation quantified as 'healing' by this measure?

The authors also discuss 'clearly defined border' of GFAP surrounding the core that is lost in IKK-CA mice. This is not clear in these images, partly due to the lines drawn over the border. Higher magnification and potentially some quantification is needed if this is to be claimed.

Figure 4

Fig. 4e is really interesting in that a chronically primed astrocytes are still responsive. They are not tolerised but primed and become much more activated with TBI.

What is even more interesting is that astrocyte activation does not tolerise microglia but again primes them. This would be the opposite to if a microglial cell was to receive exogenous stimuli, their responses likely dampened see (PMID: 29643512) and many other examples of innate immune tolerance.

One caveat here is that it is extremely difficult to quantify microglia in these traumatic lesions as many of the macrophages (microglia or monocyte-derived) will indeed be monocyte-derived cells.

It would be very interesting to know if the chronic activation of astrocytes can prime resident microglia or whether the microglia are tolerant to TBI but their increase in Iba1 is in fact due to extra recruited monocyte-derived macrophages. (see below, but I think the data here shows evidence for recruited immune cell changes, not microglia changes)

The behavioural data nicely corroborates the lesion data and is unsurprising that the animals recover to control levels considering how well animals recover/compensate motor function deficits.

Figure 5

It is very hard to understand what the take home message is here.

Most of this Figure (Figure 5 b/g/h) suggests that TBI has no additional effect on the astrocytes that already have NFκB activation (TBI IKK2-CAGFAP vs TBI control).

However, something must be driving the large lesion and behavioural changes between these groups. Indeed, supplemental Fig 3 d and e do suggest what is responsible.

The authors are right to highlight the similarity in profiles but I suggest should focus on supp data 3 d and e (move them to the main Figure) as a potential explanation to what is driving the histological lesion and behavioural changes between TBI IKK2-CAGFAP vs TBI control.

Most of the other comparisons appear to confuse this message i.e. fig 5a, c, d, e and maybe even g and h. Figures Supplemental 3d, e in supplemental three are much more compelling.

This then very nicely moves on to the flow cytometry data, Figure 6.

Figure 6

These data are very compelling, though it could be much clearer to which cell populations are being gated/quantified. The authors have gone to the trouble of sub gating different myeloid populations in supp Figure 4 but have not quantified them, this would be of interest to the reader.

Figure 6a shows major increase of myeloid cells (CD11b+ cells) but that specific gate is not shown either in the figure or the supplementals.

Figure 6b shows major increase of Ly6C+ monocytes, yet the only gate shown is for Ly6C+CD44+ cells, is this what is quantified in 6b?

Figure 6c-e shows an increase in what appear to be dendritic cells. This would be a very interesting finding. However, due to the bright staining, and what appears to be a lack of a live/dead stain in the samples, the authors would need to show either an IgG or FMO control for CD11c to show their staining is specific.

If they are DCs, what do the authors think they are doing there? Are they presenting antigen to the increased lymphoid cell populations? It would be very interesting to know the make-up of this lymphoid population considering the role that Th1/Th2 or Tregs can play in exacerbating or resolving TBI-associated injury.

The PCA plots show nice difference in CD11b cells between the key group of interest TBI IKK2-CAGFAP vs TBI control. Why do the authors then show volcano plots from TBI IKK2-CAGFAP vs sham? It would be much more beneficial for the reader to understand what are the transcripts that are changed between TBI IKK2-CAGFAP vs TBI control.

In Fig 6k and l the groups 'sham' is not even represented in the PCA plot. Is this because they sham CD11b+ cells are almost entirely microglia, whereas the TBI conditions will have many infiltrating cells? If so, is there much point comparing them in this way?

These data are extremely interesting and show that astrocyte-specific activation of NFkB drives immune cell recruitment and retention in the TBI brain. They also suggest that data in Figure 4 should be revised as it is not possible to say that these astrocyte effects in TBI are on microglia, but just as likely recruited monocyte-derived macrophages from the blood.

Figure 7

Figure 7a and b are not referred at all in the main text, if they are in the main figure they should be described. Again, however, the interesting finding is that IKK2-CAGFAP exacerbates TBI. Can the authors show this comparison instead? Those that they show are homeostatic microglia vs microglia/monocytes/DCs/neutrophils and do not inform us of what the factors are in IKK2-CAGFAP mice that are exacerbating injury.

Figure 8.

The authors want to show that tissue remodelling and scar formation are affected by astrocyte NFkB manipulation. They show that many ECM genes are downregulated in astrocytes and myeloid cells.

Then, again, the authors start to make comparisons not directly between TBI IKK2-CAGFAP vs TBI control but between TBI vs sham and TBI IKK2-CAGFAP vs sham then comparing (not directly) their outputs, for example Figure 8f. regarding an EMT gene signature. It is difficult to make conclusions this way.

To make this much stronger the authors should go back to the tissue and look at the scar formation with GFAP, col4, CSPGs or other ECM proteins are measure what the border forming capacity is. This alluded in Figure 3, but to claim scar formation and or tissue regeneration is altered, they need to show this at the tissue and protein level, on sections, ideally across time.

Methods

Much more information is needed on the model. Were animals excluded for fractures? What was the mortality rate? Were there any other exclusion criteria?

Reviewer #3

(Remarks to the Author)

This is an interesting preclinical study that uses elegant reporter and astrocytic loss of function and gain of function models to investigate NFkB activation in the context of acute TBI, wound healing and functional recovery. Overall, the preclinical studies are rigorous and well designed, and the conclusion are quite well supported by the data. However, additional evidence for deficits in wound healing in the IKK-CAGFAP model are needed, and more quantitative analysis of the glial scar dynamics would strengthen the findings of the paper.

The author should address the following points related to these new studies:

- 1) The opening figure is limited by solely focusing on bulk RNAseq data from a single timepoint (3dpi). It might be useful to add the temporal data for NFkB regulated genes (SFig 1) to the figure to highlight the dynamic response of IKK/NFkB signaling following CHI.
- 2) The use of the NFkB reporter model in Figure 2 is novel and elegant. It provides clear visualization of activation of NFkB signaling in astrocytes and microglia in the acute phase post-injury. To improve the descriptive opening of the manuscript the authors may wish to consider combining tissue level bulk RNAseq/qPCR analysis of IKK/NFkB signaling with the EYFP-reporter model for NFkB activation that details cell-specific responses in astrocytes and microglia. I believe this will lead to a stronger foundation to the research article.
- 3) The IKK2-DNGFAP and IKK2-CAGFAP loss and gain of function models for NFkB activation in astrocytes are nice. Astrocyte NFkB activation (IKK2-CAGFAP) results in lesion expansion and delayed resolution following CHI, when compared to control CHI mice. The temporal ex vivo MRI lesion assessments are interesting and support the notion of delayed wound healing response acutely following CHI, which ultimately reaches same level of injury resolution at 30 days post-injury. It is a pity that these are not in vivo longitudinal MRI assessments in the same mice because that would be more rigorous and biologically insightful. Are other DTI metrics for white matter damage across the time course available to investigate effects of IKK2-CAGFAP on TBI pathophysiology?
- 4) The glial scar in figure 3 needs to be more clearly defined and quantified in control CHI mice in order to compare lack of scar formation in IKK2-CAGFAP CHI mice. There is a robust increase in GFAP reactivity in the IKK2-CAGFAP mice under sham conditions, so it is difficult evaluate the glial scar in CHI mice. Additional quantitative measures are needed.
- 5) Glial proliferation studies in Figure 4 are interesting and indicate increased Brdu incorporation in astrocytes and microglia rapidly after CHI in the IKK2-CAGFAP mice. It would be helpful to include a schematic diagram for the region of analysis in Fig 4e and 4f. Is proliferation only occurring at the lesion boundary/glial scar, or does it also occur in contralateral regions distant from the lesion epicenter? There is a much higher level of proliferation in microglia compared with astrocytes. How do the authors interpret this early injury proliferative response and immune interactions between astrocytes and microglia?
- 6) Neurological outcomes: The CatWalk gait analysis can measure >100 specific parameters of motor function, coordination, and gait, and so it is relatively easy to show differences between groups using this analysis. How they are related to fine motor coordination impacted by TBI are less clear, and this is hotly debated in the neurotrauma field, especially when assessing motor function in injured mice. No sham mice are included in the current CatWalk analysis... Therefore, it would be important to use more established motor function tests such as accelerating rotarod and/or beam walk to examine fine motor coordination. Furthermore, cognitive function is chronically altered after CHI/TBI, so inclusion of cognitive outcomes (e.g. MWM or novel object recognition) would strengthen the analysis of post-traumatic neurological impairments in IKK2-CAGFAP mice.
- 7) RNAseq of isolated astrocytes: The important comparison of TBI IKK2-CAGFAP and TBI control has been relegated to supplemental material (SFig 3c). It would be better suited in the main figure.
- 8) NLRP3 inflammasome activation is implicated in the heightened inflammatory response in TBI IKK2-CAGFAP mice. Is this specific to microglia or astrocytes? Perhaps RNAseq data can be examined to address this open question? Also, it would be useful to analyze cleaved caspase-1, IL-1b, and IL-18 in this sample to provide a more robust analysis of NLRP3 inflammasome activation in the model.
- 9) There is a need for more concrete evidence to support deficits in wound healing in TBI IKK2-CAGFAP mice. Protein level and in situ analyses (imaging) of ECM and wound healing pathways are needed to support the transcriptomics data in Figure 8.
- 10) A much deeper analysis of osteopontin (OPN) is needed, and a rescue experiment using OPN in TBI IKK2-CAGFAP mice or inhibition of LCN2 should be considered to nail the mechanisms that impair wound healing following in TBI due to astrocytic NFkB activation (related to figure 9).
- 11) The title needs to be tempered based on existing data, and 'wound regeneration' following TBI needs to be more conclusively demonstrated with empirical evidence.

Version 1:

Reviewer comments:

Reviewer #2

(Remarks to the Author)

I thank the authors for addressing my point-by-point concerns.

I just have one final query. The way the glial scar is quantified in new Figure 4, measuring GFAP thickness around the wound core is clearly challenging. It would not be clear where to measure the core from and which 'edge' of the 'scar' to end

at. These are not nice concentric spheres and at day 3 the scar edge is almost impossible to define. The fact that this was done in a blinded manner is critical and the process would be extremely variable.

How then in Fig 4c can 'GFAP expression thickness' around the core by exactly the same measurement across three time points, with zero variability in the IKK-CA group? The lack of any variation between animals seems unlikely.

Reviewer #3

(Remarks to the Author)

NCOMMS-24-50636A: In the revised manuscript the authors addressed several of the points raised in the prior review. New data are provided using flow cytometry of various immune population in the injured brain in the genetic models, but there are concerns about the robustness of the data based on unusual gating strategies and use of % in data presentation. The revised manuscript is also very complex and is challenging to follow due to the inclusion of so much new additional data. Simplification of the major results/findings to increase focus and access for the reader is recommended.

Below are some comments about revision 1.

1. For flow cytometry studies the live/dead staining and gating strategy used for cellular populations should be included in supplemental figures for transparency and rigor.
2. Supplemental figure 4. The annotation of the statistical comparisons in the NNS and Catwalk data are confusing (some use colour, others do not...). It would be advisable to make the data more accessible to readers by improving the statistical comparisons in this figure.
3. The placement of gates in new flow cytometry studies when comparing groups of mice is problematic (Fig 8a). The same gate needs to be used in each group to investigate phenotypic changes in immune populations due to genotype.
4. The absolute numbers for immune populations in the injured brain should be presented. Are these similar to %s shown in Figure 9c-e? It is surprising that so many peripherally derived monocytes/neutrophils are being recruited so late after TBI at 30dpi. It would be informative to see the absolute numbers across the genetic models to get a clear picture of this dynamic response.
5. The studies remain highly descriptive. The mechanistic insight about wound healing are touched upon in the final figure but left open-ended. The discussion focuses on osteopontin, but this mechanisms was not examined experimentally. This is a missed opportunity to provide true mechanistic insight on role of astrocytic NfKB role in wound healing following TBI.
6. The addition of large amounts of new data in the revision means that the manuscript is extremely dense and hard to navigate for readers. This leads to a lack of focus. The authors are encouraged to simplify things and focus on critical take home messages in the results.

Version 2:

Reviewer comments:

Reviewer #2

(Remarks to the Author)

the authors have addressed my comments

Reviewer #3

(Remarks to the Author)

I thank the authors for carefully addressing R1 comments and improving the clarity of the manuscript. I am happy to approve this study for publication.

Please see below a point-by-point response to the individual comments of the referees.

Reviewer #1:

In this manuscript entitled, NFkB activation in astrocytes critically affects wound regeneration after TBI, the authors use a weight-drop model of closed-skull brain injury (in mice) to explore gain of function(GoF) and loss of function(LoF) of NFkB signaling (IKK2) in astrocytes and focus on wound healing and inflammation. This closed skull injury model induces a significant wound in the cortex of mice. Here, the authors report that TBI induces a prominent NFkB-related gene expression signature within the cortex after TBI. This corresponds with proliferation of astrocytes, cortical inflammation, and induction of a wound healing/glial scar response. Based on the initial analyses, the authors predict that astrocytic responses in NFkB signaling are important in the inflammatory response to TBI. As such, they use astrocytic-IKK2 gain of function(GoF) and astrocytic-IKK2 loss of function transgenic lines. They report that TBI-WT, TBI-CA (gain of function), and TBI-DN (loss of function) all result in different transcriptomic signatures in astrocytes. In general, the astrocytic- IKK2 gain of function mice had worse cortical lesions and increased myeloid cell infiltration compared to the WT-TBI mice. Nonetheless, the astrocytic-IKK2 (GoF) had major effects on transcriptional profiles, independent of TBI. While, astrocyte-NFkB GoF increased the size of the lesion initially (up to 15 dpi), the wound sizes were similar by 30 dpi. The dominant negative of IKK2 in astrocytes did not have much of an influence on the parameters tested. This may reduce the relevance of the data using the IKK2 overexpression in astrocytes. The overall conclusion is that over expression of IKK2 in astrocytes disrupts wound healing and this is associated with more inflammation and recruitment of myeloid cells.

1. One major concern is that in that the gain of function experiments using astrocytic-IKK2 show that there is a robust effect of genotype, independent of TBI. This effect is clear in the data from figures 4 and 5. In figure 5, the GoF in astrocytes induces a TBI-like response in the context of the transcriptional analyses. While the authors acknowledge this, it is still problematic for the study. One issue is that the gain of function of IKK2 (or overexpression) in astrocytes seems artificial (based on the results with the dominant negative version) and therefore, less biologically relevant. In addition, the study focuses only with the lesion area of the cortex after TBI. The gain of function problem in the astrocytes will be widespread throughout the brain. Furthermore, the potential confounds of this main effect of genotype is obscured throughout the paper. The authors switch from showing sham-controls to showing all TBIs groups with comparisons of the ipsilateral vs contralateral sides of the injured cortex (Figs.7-9). It is important to note that the authors show an interaction effect between astrocytic-IKK2 and TBI in the context of lesion size (up to 15 dpi), myeloid cell infiltration in the lesion, and with the transcriptional analyses. For the lesion, however, it is the same between TBI and TBI-astrocytic-IKK2 by 30 dpi. It is also unclear how the data in Fig.8 supports the author's claim on wound healing. These data are difficult to interpret and they lack sham controls. For

these reasons, the primary conclusion of the paper are not convincingly supported by the data presented.

Response: We would like to thank the reviewer for critically questioning our study and expressing the doubts about the general biological relevance of a gain-of-function approach due to the limited effects of the corresponding loss-of-function approach. To address these fundamental issues, we performed several new experiments and added now additional data to strengthen both the relevance of data derived from the gain-of-function model and to characterize the post-traumatic-consequences in the LoF model in more detail. On transcriptomic levels, the IKK2-LoF model shows an upregulation of GO terms associated with ROS protection, mitochondrial activity and metabolism (**Fig. 5g**), which might be more relevant for the long-term outcome of TBI. On cellular level, we found a faster clearance of CD11c+ and lymphoid cells in the later stages of TBI (**Supplementary Fig. 10i/j**), while immunoblot analyses showed persistently elevated levels of osteopontin, a factor known to improve TBI outcome (**Supplementary Fig. 11i**). There is also some reduction in microglia and astrocyte proliferation (**Figure 6**) in the LoF model, which however did not significantly affect the overall wound healing process.

We now also performed longitudinal MRI analysis until 30 dpi to follow lesion size development in the same animal over time in more detail validating the deficient wound healing process in the GoF model, showing a significant difference at day 15 and day 30 (**Figure 3**).

Regarding the overt relevance of the IKK2-CA^{GFAP} model, it became clear in the last years that IKK/ NF- κ B signaling increases with advancing age and in the context of neurodegenerative disorders such as AD or PD (Kaltschmidt et al., 2022, PMID: 35983068). As the incidence of TBI increases in the elderly, TBI in the context of elevated NF- κ B activity is a relevant clinical scenario affecting TBI outcome thus supporting the biological and clinical relevance of the GoF model for the utilization in preclinical TBI research. Accordingly, our IKK2-CA^{GFAP} mice clearly recapitulate features usually observed in mice of advanced age, such as polarization to an astrocytic A1 phenotype (**Supplementary Fig. 7d/e**), expression of SASP factors (**Supplementary Fig. 7f**), reactive gliosis (**Figure 6**) as well as a profound leukocyte infiltration (**Figure 8**) as indicated by the GoF model in the non-traumatized state.

Concerning the wound healing deficit in IKK2-CA^{GFAP} mice (depicted and referenced in **Figures 3, 4, 6, 7 and 10**), we examined the transcriptomic profile of both ACSA2+ astrocytes and CD11b+ myeloid/microglia cells at day 3 post TBI, and found a significant downregulation of genes which are associated with tissue remodeling/ECM organization in the GoF model (**Figure 7; Supplementary Fig. 6f**). This illustrates the prominent deficit in the regulation of genes involved in wound healing after TBI in IKK2-CA^{GFAP} animals already at an early time point. We also included Sham controls as indicated (**Supplementary Fig. 11i**).

Altogether, this suggests that the GoF model recapitulates aging-related deficits in astrocyte function, including impaired wound healing in response to TBI.

2. Another related concern is that astrocytic-IKK2 dominant negative (IKK2-DN^{GFAP}) does not affect TBI induced CNS scar formation or myeloid cell infiltration. The transcriptional effects in the astrocytic-IKK2, seem different (from the PCA plot) but were not explored in the manuscript. Based on the data in Fig.1 and the assumptions of the introduction, the astrocytic-IKK2 dominant negative should reduce the cortical inflammation after TBI, but it does not. Because it does not seem to have much of a role normally, the overexpression of IKK2 in astrocytes has reduced biological significance.

Response: We understand the reviewer's concerns regarding the obvious lack of consequences on TBI induced CNS scar formation or myeloid cell infiltration in the IKK2-DN^{GFAP} model. Therefore, we performed additional experiments and following the reviewer's suggestion also explored the transcriptional profile of the IKK2-DN^{GFAP} model in isolated cells and in bulk tissue in more detail (**Supplementary Data 1; Supplementary Fig. 3**). Overall, this reveals i) selective suppression in inflammatory gene expression and ii) pronounced changes in genes and processes linked to ROS protection and mitochondrial processes as indicated by the enrichment chart in **Figure 5g**. In addition, western blot analysis revealed upregulation of osteopontin at 3 dpi (**Supplementary Fig. 11a/c**), which was still detectable at 14 dpi in the IKK2-DN^{GFAP} model but not in controls. (**Supplementary Fig. 11i**). Furthermore, the infiltration of peripheral immune cells was resolved faster as shown in **Supplementary Fig. 10i/j**.

3. There are also some concerns on novelty and relevance of the study. For example, it is well established that TBI causes inflammation and increases in NFkB mediated pathways in the brain of rodents. Moreover, myriad neurotrauma (SCI and TBI) studies in rodents show that astrocytes are critical for glia scar formation, are essential for wound healing, and influence immune cell infiltration in the brain and spinal cord. Here, the main findings is that a gain of

function/overexpression of NF κ B signaling in astrocytes impairs the wound healing response at the lesion site.

Response: We agree with the reviewer that NF- κ B activation following traumatic brain injury is long established, and it is also well-studied that astrocytes are the cells critical for glia scar formation, they are essential for wound healing, and they are able to influence immune cell infiltration. However, to our knowledge there is no genetic evidence so far that NF- κ B activation in astrocytes is functionally relevant for scar formation/wound healing in TBI. Furthermore, the specific role of NF- κ B in the post-traumatic remodeling/infiltration process has not been well investigated as of now. Thus, characterization of the mechanistic consequences of NF- κ B activation and inhibition in astrocytes for the regulation of TBI outcome is not established and novel. We also show here that astrocyte-specific activation of IKK/ NF- κ B signaling is sufficient to phenocopy aging-associated changes including age-dependent infiltration of peripheral immune cells. As a consequence, specific pre-existing astrocyte activation/reactivity conditions are created which impair the ability of astrocytes to limit tissue damage and to promote repair thus recapitulating wound healing deficiencies observed in the aged brain. Our findings also support the view that IKK/ NF- κ B-mediated pre-activation of astrocytes gives rise to paracrine signaling that promotes global neuroinflammation possibly “overwriting” the local signals required for instructing the localized protective scar formation finally worsening TBI outcome. However, selective repression of NF- κ B in astrocytes using IKK2-DN expression seems not to be sufficient to efficiently counteract TBI-induced global neuroinflammation and to provide conditions which result in an obvious benefit for the CNS scar formation process.

4. Another concern is that the description for the generation of the dominant negative (LoF) and constitutively activated (GoF) cell specific mouse lines are not included within the manuscript. Also, the authors don't provide any validation that the NF κ B pathway is overexpressed in astrocytes and that the NF κ B pathway is non-functional in astrocytes. Validation of the models is an important element of rigor and reproducibility.

Response: We agree with the reviewer that validation of the models is an important element of rigor and reproducibility. For this, we now added additional information on the generation of the conditional mouse models in the method section, and importantly

included additional data to demonstrate the continued functionality of our animal models (**Supplementary Fig. 2/3**).

Transgene expression in the GoF and LoF model is coupled to a luciferase reporter gene and robust luciferase activity was validated by IVIS prior to TBI and protein assay approaches (**Supplementary Fig. 2a/3a**) showing that postnatal doxycycline withdrawal results in robust IKK2-CA/IKK2-DN transgene expression. As a consequence, upregulation of the IKK/ NF- κ B pathway in the GoF model induces an elevated baseline expression of several NF- κ B target genes, which we show in cortical and cerebellar tissue. For IKK2-CA transgene function on protein level, we would also like to refer to the increased presence of phospho-p65 levels, which are indicative for an activated IKK/ NF- κ B pathway (**Supplementary Fig. 2e/f**).

Furthermore, we isolated primary astrocytes from adult GoF and LoF animals, took them in culture and performed gene expression analyses by qRT-PCR. Here, basal NF- κ B dependent gene expression is significantly elevated in the GoF model whereas these genes are repressed in the LoF model. When we induced NF- κ B in LoF astrocytes by the application of a stimulation cocktail consisting of LPS/TNF/Poly I:C, we observed a prominent repression but not a complete blockade of NF- κ B target gene expression. This indicates, that the dominant-negative IKK2-DN allele is functional but it is not capable to completely render the NF- κ B pathway non-functional in IKK2-DN^{GFAP} astrocytes (**Supplementary Fig. 3d**).

5. In the assessment BrdU+ IBA1 and GFAP at 3dpi, there are no sham groups included.

Response: We thank the Reviewer for pointing out the lack of sham controls. We now included the respective sham controls, which show a similar behavior to the day 1 cohort in **Figure 6**.

6. In Fig.2 the authors only show representative images at 3 dpi. Authors should show all time points, not just 3dpi, especially because it seems to peak at 7dpi.

Response: We thank the reviewer for this comment and therefore included additional data to this part in the revised manuscript. Using a flow cytometric approach, we also characterized the EGFP/ NF- κ B activity up to day 30 post TBI. Both methods revealed similar NF- κ B activation kinetics for astrocytes and microglia concerning day 3 and

day 7. Furthermore, we now also monitored the post-traumatic NF- κ B activation kinetic in O4+ oligodendrocytes (**Figure 2h**) as well as several leukocyte subpopulations (**Figure 2i/j**; **Supplementary Fig. 1g-i**).

7. Concern that in Fig.3 the text does not match the data/representative images. Authors state that controls increase from day 1 to day 3, but it seems to decrease in the image and not really change in the graph. As they say, the lesion decreases in controls from day 3 to day 7 – 15 but increases and then stays the same in the GoF. In Fig.3b, there is a key for all three groups but then they do not include the LoF group. It also looks like the representative image do not match the graph for 1dpi. It is also unclear why Fig.3c separates all the groups and what time point this graph represents

Response: We thank the reviewer for pointing out this issue and included a quantification and representative images of the LoF model in **Figure 3a/b**. Furthermore, we changed several representative images of the TBI GoF wound lesion area to highlight the disturbed glial scar formation in **Figure 3a/b**. In addition, we performed now longitudinal MRI analysis by 30 dpi to follow lesion size development in the same animal over time in more detail as depicted in **Figure 3c/d**.

8. Fig. 6. shows volcano plots for TBI-WT v sham and TBI-CA v sham but not direct comparison, even though that is the group of most interest.

Response: We thank the reviewer for this suggestion and included the direct comparison in the main **Figure 5d/e**.

9. Concern that the data in Fig.7 are disjointed. It starts with data from fig 6 and then moves to blots of ipsi/contra tissue and then ending with more astrocyte transcriptional data from before. It is unclear if control is contralateral tissue and CA is ipsi or if this is just a genotype comparison. Also unclear why in Fig.7d-f, it switches from including sham controls to contralateral controls.

Response: We thank the review for pointing out the confusing description and rearranged **Figure 7**. We removed **Figure 7a,b,g** and **i** due to redundancy and moved **Figure 7h** to the new **Figure 7** as **Figure 7g**. The new **Figure 7**, designated as **Figure 9** is now characterizing the post-traumatic immune response and inflammasome activity. Regarding the western blot data, we decided to include the contralateral tissue (co) as an internal control for both control (Ctrl) and IKK2-CA^{GFAP} animals, as protein

expression in non-lesioned tissue did not show any differences to the sham cohort at this time point. See also a representative sham cohort in **Supplementary Fig. 11i**.

10. Fig. 9 shows western blots of proteins they say are important for wound healing and inflammation. Nonetheless, no citations supporting this idea are provided. The author's use of the ratio of osteopontin (protective) to lipocalin-2 (inflammatory) is arbitrary.

Response: We thank the reviewer for this note and included now the corresponding references to this issue in the result section and not only in the discussion part. Overall, we also further elaborate on the protective role of osteopontin as well as the detrimental findings on lipocalin-2 in the discussion section.

Reviewer #2:

In the submitted manuscript, Hein et al use astrocyte-specific strategies to modulate NfKB activation through loss of function (IKK2-DN GFAP) or gain of function (IKK2-CA GFAP) mouse models and investigate how this affects various outcomes in a closed head injury model of TBI. An important role for NfKB following traumatic injury is long established, though the attempt to look at this from an astrocyte-specific perspective is interesting and novel. A large proportion of the data in the paper is from bulk transcriptomic analysis of either whole tissue or isolated cell populations. Much of this can be confusing as direct comparisons are often made between groups that are of secondary interest and then descriptive comparisons of these comparisons are made (see below). The main finding of the paper is that IKK2-CA GFAP exacerbates many readouts after TBI including behavior, tissue lesion size and, most interestingly, immune cell recruitment.

The title of the paper talks about 'wound regeneration' but the paper doesn't really get into what 'regeneration' means and its data to show actual wound regeneration is lacking. It would be good if the authors could clarify what they mean by 'wound healing' and 'wound closure' and what this means in the brain? Also, a major theme of the paper is scar formation and tissue remodeling, and there is transcriptional and some protein analysis to support this, though a much more thorough analysis of the scar and tissue in tissue sections would be needed to support this (see below).

The major interesting finding in this paper is that astrocyte-specific activation of NfKB is sufficient to massively induce a peripheral infiltration of immune cells that are likely responsible for increased tissue damage and lack of resolution of inflammation. This has not been shown before and is of interest to the field.

Lastly, the authors largely focus on gain of function (IKK2-CAGFAP) rather than loss of function IKK2-DN GFAP as the latter had little to no effect on the model. How does this fit with the authors' view that NfKB could be an effective therapeutic target?

I have multiple comments that would also need addressing, in figure order:

Figure 2

Figure 2e authors describe microglia maintaining NfKB expression but an alternative explanation could be that Iba1 positive recruited monocyte-derived macrophages are taking their place. The authors need to do experiments to show definitively microglia are changing, or tone down microglial-specific conclusions

Response: We thank the reviewer for mentioning this very valid concern. To get a more comprehensive view in the post-traumatic NF- κ B activation kinetics and the participating players, we included now detailed flow cytometry analyses and stained cells obtained from the insult area of kappa-EGFP reporter mice post TBI for several leukocytic subsets as depicted in **Figure 2** and **Supplementary Fig. 1**. We defined microglia as CD45^{low}CD11b⁺ and confirmed the EGFP expression kinetics observed in the histological sections (**Figure 2j**; **Supplementary Fig. 1g**). Co-staining with CD44

in previous experiments and also in the new infiltration data showed a strong staining for the CD45^{high} population which can be used as a marker of cells with peripheral origin (**Supplementary Fig. S10c**) as previously described in literature (Korin *et al.*, 2017; PMID: 28758994). This CD45^{high}CD11b+CD44+ cell population, which includes monocytes and macrophages, did not show a major contribution to the post-traumatic NF-κB activation pattern, only exhibiting a more noticeable upregulation for the later stages. In contrast to the histological analysis, flow cytometry showed a baseline expression of NF-κB for the CD45^{low}CD11b microglia, which visibly increases until 7dpi. These additional stainings indicate, that the CD45^{low} myeloid cells are the main contributors of the EGFP signal and, therefore, further support our initial statement concerning the post-traumatic NF-κB activation in microglia (**Figure 2j**)

Figure 3

Fig. 3b measures 'Trauma dimension' and 3c is 'TBI area'. Fig 3c certainly is underpowered if the authors want to claim a no change.

Response: We thank the reviewer for pointing out this inconsistency. We now rearranged the data in **Figure 3** and included a quantification of the lesion area in mm² as 'trauma dimension' for the IKK2-DN^{GFAP} model until 15 dpi. While we certainly agree with the Reviewer's concern regarding the statistical power, the follow-up of the scar formation process in the IKK2-DN^{GFAP} model showed no significant differences to the control cohort in neither the histological sections nor in the new longitudinal MRI analysis. Therefore, we conclude that downregulation of the IKK/ NF-κB pathway in astrocytes has no significant effects on the parameters we did measure for lesion progression over time.

Fig. 3c shows control vs IKK2-DN^{GFAP} but this is not referred to in the text at all.

Response: We have now corrected this mistake and included the description of the findings made in the IKK2-DN^{GFAP} model in the result part.

Fig. 3d How do the authors account for what appears to be other lesion sites across the brain when quantifying lesion volume? Ex vivo imaging clearly shows major cavities proximal and distal to the arrow to where they indicate lesion was measured.

Response: The cavities seen in the former **Figure 3d** are artifacts. For ex-vivo measurement, the fixed brains are transferred into Fluorinert solution. This solution should be free of bubbles as these may interfere with the MRI measurements. In case of the image shown for day 30, there were almost certainly some bubbles present, as the Fluorinert solution slightly changes its consistency in dependence of the ambient temperature. However, for better clarity, we now removed the previous *ex-vivo* MRI results from our study and, in turn, included now the results of a longitudinal *in-vivo* study for the same time points shown **Figure 3c/d**.

Figure 4

Fig. 4e is really interesting in that a chronically primed astrocytes are still responsive. They are not tolerized but primed and become much more activated with TBI.

What is even more interesting is that astrocyte activation does not tolerize microglia but again primes them. This would be the opposite to if a microglial cell was to receive exogenous stimuli, their responses likely dampened see (PMID: 29643512) and many other examples of innate immune tolerance. One caveat here is that it is extremely difficult to quantify microglia in these traumatic lesions as many of the macrophages (microglia or monocyte-derived) will indeed be monocyte-derived cells.

It would be very interesting to know if the chronic activation of astrocytes can prime resident microglia or whether the microglia are tolerant to TBI but their increase in Iba1 is in fact due to extra recruited monocyte-derived macrophages. (see below, but I think the data here shows evidence for recruited immune cell changes, not microglia changes)

Response: We agree with the reviewer's concern that infiltrating monocytes and macrophages might dilute the TBI effect on microglia. Classification of microglia and peripheral infiltrates was conducted by evaluation of the IBA1 intensity, as macrophages were defined as IBA1^{dim}. However, to strengthen our data, we conducted additional flow-cytometric analysis, as this method allows a better and clearer resolution of receptor expression densities (**Figure 2j; Supplementary Fig. 1**). We identified peripheral infiltrates as CD45^{high}CD11b⁺CD44⁺, as expression of CD44 has been associated with leukocytic migration into the CNS (Korin *et al.*, 2017; PMID: 28758994; Böttcher *et al.*, 2019; PMID: 30559476). Co-staining with the

microglial marker P2RY12 also showed an absence on the CD45^{high} myeloid population. According to this gating strategy, myeloid cells with high CD45 expression density were identified as peripheral infiltrates. However, while we agree with the reviewer's observation that many of the neuroinflammatory effects found in this study are caused by peripheral infiltrates, we also saw an upregulation of CD11c and MHC class II in CNS-resident CD45^{low}CD11b⁺CD44⁻ microglia in the GoF mouse model (**Supplementary Fig. 10b**). This finding also implies a paracrine effect of NF- κ B - activated astrocytes which may lead to a specific activation status of microglia cells.

Figure 5

It is very hard to understand what the take home message is here. Most of this Figure (Figure 5 b/g/h) suggests that TBI has no additional effect on the astrocytes that already have NF κ B activation (TBI IKK2-CAGFAP vs TBI control). However, something must be driving the large lesion and behavioural changes between these groups. Indeed, supplemental Fig 3 d and e do suggest what is responsible.

The authors are right to highlight the similarity in profiles but I suggest should focus on supp data 3 d and e (move them to the main Figure) as a potential explanation to what is driving the histological lesion and behavioral changes between TBI IKK2-CAGFAP vs TBI control. Most of the other comparisons appear to confuse this message i.e. fig 5a, c, d, e and maybe even g and h. Figures Supplemental 3d, e in supplemental three are much more compelling.

Response: We appreciate the reviewer's suggestion for improvement and implemented this suggestion. We now rearranged the figure and main text accordingly to convey our message more clearly. The data are now presented in **Figure 5** that depicts the PCA (**Figure 5c**) as well as volcano plots (**Figure 5d/e**) and enrichment charts (**Figure 5f/g**) directly comparing TBI IKK2-CA^{GFAP} or TBI IKK2-DN^{GFAP} to TBI Control. We removed the original comparisons to the Sham control group or moved them into the Supplement (**Supplementary Fig. 6**), as we agree completely with the Reviewer's assessment.

Figure 6

These data are very compelling, though it could be much clearer to which cell populations are being gated/quantified. The authors have gone to the trouble of sub gating different myeloid populations in supp Figure 4 but have not quantified them, this would be of interest to the reader.

Response: We appreciate the reviewer's comment and thank for the suggestion. We now included the quantification of markers expressed in the CD11c+ population of IKK2-CA^{GFAP} animals in **Supplementary Fig. 9c**. Furthermore, we also present the gating strategies of several leukocytic subsets in **Figure 8a/b**, **Supplementary Fig. 9a/b** and **Supplementary Fig. 10a**.

Figure 6a shows major increase of myeloid cells (CD11b+ cells) but that specific gate is not shown either in the figure or the supplementals.

Response: We thank the reviewer for pointing out the missing gating strategy and included this information in **Figure 9a**.

Figure 6b shows major increase of Ly6C+ monocytes, yet the only gate shown is for Ly6C+CD44+ cells, is this what is quantified in 6b?

Response: We thank the reviewer for this suggestion and included the respective gating strategies. All of our CD45^{high} myeloid cells do co-express CD44 as depicted in **Supplementary Fig. 10c**. To make the qualitative composition of the CD45^{high} infiltrates clearer and more comprehensive, we now also included a diagram showing the proportion of the populations relative to the total infiltration found at day 3 post TBI in controls, LoF and GoF animal models as shown in the main **Figure 8f**.

Figure 6c-e shows an increase in what appear to be dendritic cells. This would be a very interesting finding. However, due to the bright staining, and what appears to be a lack of a live/dead stain in the samples, the authors would need to show either an IgG or FMO control for CD11c to show their staining is specific.

Response: We conducted a live/dead staining using DAPI, and we also included Isotype controls for our gating. A representative image is now depicted in **Supplementary Fig. 9b**.

If they are DCs, what do the authors think they are doing there? Are they presenting antigen to the increased lymphoid cell populations? It would be very interesting to know the make-up of this lymphoid population considering the role that Th1/Th2 or Tregs can play in exacerbating or resolving TBI-associated injury.

Response: We appreciate the reviewer's interest in the dendritic cell data and on their role in the regulation of the TBI outcome (former **Figure 6**, now **Figure 8**). Overall, the

role of dendritic cells in TBI is not intensively studied so far. We observed a strong preference of CD11c⁺ immune cell infiltration in the lesioned brain area of IKK2-CA^{GFAP} animals. Subtyping of this cell population revealed a quite heterogeneous make-up of CD11c⁺/MHC class II^{high}/CD44⁺/Ly6C^{lo/hi} cells probably including both conventional (cDC) and monocyte-derived (moDC) dendritic cells (DCs). In this context, the enrichment of IFN- γ -associated pathways and elevated *Csf2* expression in the myeloid transcriptome induced by NF- κ B activation in astrocytes may create conditions similar to EAE promoting monocyte to moDC differentiation (e.g. Mundt et al., 2022; PMID: 36327896) and may explain elevated DC levels already under sham conditions in the IKK2-CA^{GFAP} model. As these cells can terminally differentiate into pro-inflammatory macrophages (Amann *et al.*, 2023; PMID: 36759712), they may further aggravate the neuroinflammatory response in GoF animals.

Concerning the lymphoid compartment, we included now additional data in **Supplementary Figure 10e-g**. We found that frequencies of CD4 and CD8 cells are significantly different between control and IKK2-CA^{GFAP} animals at 3 days post TBI, exhibiting a preference for CD8⁺ CTLs in the GoF model. Interestingly, these differences in ratio are equalized at 14 days post TBI, showing a significant increase in CD8a⁺ and a subsequent decrease in CD4⁺ T Cell frequencies, similar to the situation in the GoF model at 3 dpi. B220⁺MHCII⁺ cells, which we assigned as B cells, did not show any significant differences between control and GoF. The increased frequency of CD8a⁺ T cells may aggravate post-traumatic tissue damage, as these cells can be potentially activated by DCs and may trigger inflammatory neurodegeneration as well as neurological manifestations after TBI. We have now further elaborated on this issue and on the role of DCs in mounting autoimmune responses in the discussion part.

The PCA plots show nice difference in CD11b cells between the key group of interest TBI IKK2-CAGFAP vs TBI control. Why do the authors then show volcano plots from TBI IKK2-CAGFAP vs sham? It would be much more beneficial for the reader to understand what are the transcripts that are changed between TBI IKK2-CAGFAP vs TBI control. In Fig 6k and l the groups 'sham' is not even represented in the PCA plot. Is this because they sham CD11b+ cells are almost entirely microglia, whereas the TBI conditions will have many infiltrating cells? If so, is there much point comparing them in this way?

These data are extremely interesting and show that astrocyte-specific activation of NFkB drives immune cell recruitment and retention in the TBI brain. They also suggest that data in Figure 4 should be revised as it is not possible to say that these astrocyte effects in TBI are on microglia, but just as likely recruited monocyte-derived macrophages from the blood

Response: The reviewer's concerns are very valid and therefore we removed the sham comparisons and only compare the TBI groups directly to each other to further highlight the transcriptomic changes between the different genotype groups.

We also agree with the reviewer's observation that several changes are due to the presence of blood-derived leukocytes, however we could also identify changes in the microglia population (CD45^{low}CD11b+CD44-) themselves, such as an upregulation of MHC class II and CD11c receptor expression as shown in **Supplementary Fig. 10b**.

Figure 7

Figure 7a and b are not referred at all in the main text, if they are in the main figure they should be described. Again, however, the interesting finding is that IKK2-CAGFAP exacerbates TBI. Can the authors show this comparison instead? Those that they show are homeostatic microglia vs microglia/monocytes/DCs/neutrophils and do not inform us of what the factors are in IKK2-CAGFAP mice that are exacerbating injury.

Response: We thank the reviewer for pointing out this important issue. We have updated now all of the figure references and reorganized the paper as suggested. We further removed the sham comparison and moved the comparison between TBI Control vs TBI IKK2-CA^{GFAP} into the main **Figure 8g/h** and **9a**.

Figure 8.

The authors want to show that tissue remodelling and scar formation are affected by astrocyte NFkB manipulation. They show that many ECM genes are downregulated in astrocytes and myeloid cells.

Then, again, the authors start to make comparisons not directly between TBI IKK2-CAGFAP vs TBI control but between TBI vs sham and TBI IKK2-CAGFAP vs sham then comparing (not directly) their outputs, for example Figure 8f. regarding a an EMT gene signature. It is difficult to make conclusions this way.

To make this much stronger the authors should go back to the tissue and look at the scar formation with GFAP, col4, CSPGs or other ECM proteins are measure what the border forming capacity is. This alluded in Figure 3, but to claim scar formation and or tissue regeneration is altered, they need to show this at the tissue and protein level, on sections, ideally across time.

Response: We thank the reviewer for this suggestion and rearranged the figure accordingly. However, while we agree that direct comparison between TBI GoF and TBI controls is a more suitable approach and therefore also now included the corresponding GSEA plot (**Figure 7b**), we decided to not remove the comparison against Sham for the EMT in order to emphasize the inability of the GoF astrocytes to adopt this type of transcriptomic signature.

We also conducted and included more thorough analyses of factors involved in the repair and remodeling of injured brain tissue. To further investigate the scar formation on a cellular level, we included fibronectin staining over the time of 30 days – as shown in the **Supplementary Fig. 5b/c**, as well as a deeper characterization of the glial scar structure and thickness seen in the new **Figure 4**. In this context, the GFAP scar is quantified using the intensity of the GFAP staining around the wound core, while the GFAP thickness describes the distance from the TBI lesion core until the first signal was observed. Furthermore, we also stained for NG2/CSGP4 to evaluate the deposition around the lesion core as well as the reactivity of the NG2 glia. While reactivity was unchanged, deposition was significantly lower in the IKK2-GoF cohort at day 7 post TBI.

Methods

Much more information is needed on the model. Were animals excluded for fractures? What was the mortality rate? Were there any other exclusion criteria?

We have now added the relevant information and expanded upon it in our method section. Mortality of the mice was quite low (~5%) due to our optimized post-operational care. Due to the experimental procedure, the generation of a relevant insult was estimated right after trauma induction due to formation and extent of visible bleedings beneath the exposed skull. We therefore excluded animals without immediately visible or very minor lesions. Fractures– if present – were usually superficial and animals with fractures were kept in the analysis.

Reviewer #3:

This is an interesting preclinical study that uses elegant reporter and astrocytic loss of function and gain of function models to investigate NFκB activation in the context of acute TBI, wound healing and functional recovery. Overall, the preclinical studies are rigorous and well designed, and the conclusion are quite well supported by the data. However, additional evidence for deficits in wound healing in the IKK-CAGFAP model are needed, and more quantitative analysis of the glial scar dynamics would strengthen the findings of the paper.

The author should address the following points related to these new studies:

1) The opening figure is limited by solely focusing on bulk RNAseq data from a single timepoint (3dpi). It might be useful to add the temporal data for NFκB regulated genes (SFig 1) to the figure to highlight the dynamic response of IKK/NFκB signaling following CHI.

Response: We thank the reviewer for this suggestion and rearranged **Figure 1** accordingly. The longitudinal expression kinetic of selected NF-κB regulated genes measured by qRT-PCR is now assigned as **Figure 1f-m**.

2) The use of the NFκB reporter model in Figure 2 is novel and elegant. It provides clear visualization of activation of NFκB signaling in astrocytes and microglia in the acute phase post-injury. To improve the descriptive opening of the manuscript the authors may wish to consider combining tissue level bulk RNAseq/qPCR analysis of IKK/NFκB signaling with the EYFP-reporter model for NFκB activation that details cell-specific responses in astrocytes and microglia. I believe this will lead to a stronger foundation to the research article.

Response: We appreciate the remark of the reviewer and rearranged **Figure 1** and **Figure 2** as we conducted now a more thorough analysis of the post-traumatic NF-κB activation kinetics using the NF-κB reporter model. To get detailed information on the identity of the cell populations responsive to TBI-mediated NF-κB induction, we now performed flow cytometric analysis at day 1, 3, 7, 14 and 30 post TBI and stained for astrocytes (ACSA-2), oligodendrocytes (O4) (**Figure 2g-i**) and leukocytic subsets including resident microglia (CD45^{low}CD11b+CD44-), myeloid (CD45^{high}CD11b+CD44+) and lymphoid (CD45^{high}CD11b-) infiltrates (**Figure 2j**, **Supplementary Fig. 1g-i**). This approach also allowed as to confirm microglia as main contributors of the post-traumatic IKK/NF-κB activation among the myeloid cells.

3) The IKK2-DNGFAP and IKK2-CAGFAP loss and gain of function models for NFkB activation in astrocytes are nice. Astrocyte NFkB activation (IKK2-CAGFAP) results in lesion expansion and delayed resolution following CHI, when compared to control CHI mice. The temporal ex vivo MRI lesion assessments are interesting and support the notion of delayed wound healing response acutely following CHI, which ultimately reaches same level of injury resolution at 30 days post-injury. It is a pity that these are not in vivo longitudinal MRI assessments in the same mice because that would be more rigorous and biologically insightful. Are other DTI metrics for white matter damage across the time course available to investigate effects of IKK2-CAGFAP on TBI pathophysiology?

Response: We appreciated the reviewer's comment and suggestion. To strengthen our data, we now performed a longitudinal in-vivo MRI (T2-weighted) analysis of control, IKK2-CA^{GFAP} and IKK2-DN^{GFAP} at day 1, 15 and 30 post TBI, which we included as Figure 3c/d in replacement of the previous ex-vivo MRI data. These results further support our findings, clearly showing a deficit in the post-traumatic wound healing in the GoF model that was significantly deficient at day 15 and still impaired by tendency at day 30 in comparison to the control cohort. Quantification of the IKK2-DN^{GFAP} model showed no obvious differences in the wound lesion progression. Unfortunately, we could not determine additional DTI metrics.

4) The glial scar in figure 3 needs to be more clearly defined and quantified in control CHI mice in order to compare lack of scar formation in IKK2-CAGFAP CHI mice. There is a robust increase in GFAP reactivity in the IKK2-CAGFAP mice under sham conditions, so it is difficult evaluate the glial scar in CHI mice. Additional quantitative measures are needed.

Response: We agree with the reviewer's concerns and performed additional quantitative measures of the scar area. We replaced the original images with ones depicting a clearer and better-defined glial limitans border, also including data from the IKK2-DN^{GFAP} model (**Figure 3a/b**). The quantitative measures revealed a decreased deposition of NG2/CSPG4, an unchanged reactivity of NG2 glia (determined by hypertrophy), as well as a decreased expression but - at day 7 post TBI - unaffected thickness of GFAP area (**Figure 4**) in the IKK2-CA^{GFAP} model. We also identified significantly broader expression of fibronectin in the GoF model (**Supplementary Fig. 5**) overall validating the aberrant scar formation in these animals.

5) Glial proliferation studies in Figure 4 are interesting and indicate increased BrdU incorporation in astrocytes and microglia rapidly after CHI in the IKK2-CAGFAP mice. It would be helpful to include a schematic diagram for the region of analysis in Fig 4e and 4f. Is proliferation only occurring at the lesion boundary/glial scar, or does it also occur in contralateral regions distant from the lesion epicenter? There is a much higher level of proliferation in microglia compared with astrocytes. How do the authors interpret this early injury proliferative response and immune interactions between astrocytes and microglia?

Response: We now included a schematic diagram concerning the region of analysis in the TBI insult area (**Figure 6a**) as well as explanations regarding the approach of quantification (**Supplementary Fig. 5a**). In the control and LoF cohort, proliferation and gliosis was restricted to the lesion area while in the GoF group, baseline astrogliosis – but not proliferation - was present in the whole brain (**Figure 6c**). The higher level of proliferation in IBA1+ microglia may be in parts due to infiltrating monocytic/dendritic cells, which also stain positive for IBA1 (Zhang *et al.*, 2021; PMID: 33582751). In line with this, microglia as the resident immune cells of the brain parenchyma, do react strongly to DAMPs and PAMPs, and therefore may show a stronger proliferation in comparison to other resident cell types.

6) Neurological outcomes: The CatWalk gait analysis can measure >100 specific parameters of motor function, coordination, and gait, and so it is relatively easy to show differences between groups using this analysis. How they are related to fine motor coordination impacted by TBI are less clear, and this is hotly debated in the neurotrauma field, especially when assessing motor function in injured mice. No sham mice are included in the current CatWalk analysis. Therefore, it would be important to use more established motor function tests such as accelerating rotarod and/or beam walk to examine fine motor coordination. Furthermore, cognitive function is chronically altered after CHI/TBI, so inclusion of cognitive outcomes (e.g. MWM or novel object recognition) would strengthen the analysis of post-traumatic neurological impairments in IKK2-CAGFAP mice.

Response: The CatWalk system has been developed to automatically and in particular observer independently assess gait and motor function in rodents. It has been used in a variety of preclinical models of traumatic and non-traumatic neurologic conditions such as stroke, peripheral nerve injury, spinal cord injury as well as traumatic brain injury.

We agree with the Reviewer's concerns that it is less clear how the >100 specific parameters of motor function, coordination, and gait recorded by the CatWalk system

are exactly related to TBI-induced deficits in motor performance. However, there are several parameters for instance the mean of max intensity at (% relative to stand time) that were consistently reported as relevant in TBI and stroke models, implying that they could potentially serve as main alerting parameters of gait dysfunction in these models (PMID: 37351154). We also included now sham mice in the CatWalk analysis to strengthen the impact of TBI on motor coordination.

Furthermore, we included a neurological severity scoring (**Supplementary Fig. 4b**) which includes beam walking and beam/round stick balancing. The NSS revealed an increase in neurological impairments 6 h post TBI in both control littermates (median NSS = 5) and IKK2-CA^{GFAP} mice (median NSS = 6) compared to corresponding sham animals (median NSS = 2 and 2.25, respectively). These impairments mainly include motor dysfunction during beam walking and beam/round stick balancing. After 24 h, the NSS is already decreasing again in both controls and IKK2-CA^{GFAP}, however, the deficits of IKK2-CA^{GFAP} mice are significantly increased compared to controls at this time point and also present until day 3 post TBI. Later on, no significant differences can be detected between groups. Therefore, both CatWalk and NSS analysis indicate a transient deficit in motor performance in the GoF model.

We agree that the inclusion of cognitive outcomes (e.g., MWM or novel object recognition) would strengthen the analysis of post-traumatic neurological impairments in IKK2-CA^{GFAP} mice. However, this type of behavioral tests needs a comprehensive set of new experiments, an issue that is not manageable in this research study, in particular as these tests need lengthy additional animal testing approvals.

7) RNAseq of isolated astrocytes: The important comparison of TBI IKK2-CA^{GFAP} and TBI control has been relegated to supplemental material (SFig 3c). It would be better suited in the main figure.

Response: We thank the reviewer for this remark and moved the TBI IKK2-CA^{GFAP} vs TBI control comparison to the main figure (**Figure 5d-g**). We removed most of the sham comparisons or moved them into the **Supplementary Fig. 6**, as they do not contribute critically to the overall message and conclusion of our study. Expression of DEGs and enrichment plots in comparison to the Sham Control cohort can now be found in **Supplementary Fig. 6a-d**.

8) NLRP3 inflammasome activation is implicated in the heightened inflammatory response in TBI IKK2-CAGFAP mice. Is this specific to microglia or astrocytes? Perhaps RNAseq data can be examined to address this open question? Also, it would be useful to analyze cleaved caspase-1, IL-1b, and IL-18 in this sample to provide a more robust analysis of NLRP3 inflammasome activation in the model.

Response: Following the Reviewer's suggestion, examination of the RNAseq data revealed a significant upregulation of NLRP3 inflammasome components in astrocytes, whereas no major differences could be observed in CD11b+ cells. This suggests that changes in inflammasome component expression is rather restricted to astrocytes (**Figure 9j**).

9) There is a need for more concrete evidence to support deficits in wound healing in TBI IKK2-CAGFAP mice. Protein level and in situ analyses (imaging) of ECM and wound healing pathways are needed to support the transcriptomics data in Figure 8.

Response: We thank the reviewer for the comment and added more data to support the deficits in wound healing. For this, we performed deeper analyses of the lesion area with additional quantitative measures including deposition of NG2/CSPG4, reactivity of NG2 glia, expression and thickness of GFAP (**Figure 4**) as well as expression of fibronectin (**Supplementary Fig. 5**). On gene expression levels, we added the expression kinetics of several ECM/scar-associated factors until day 30 post TBI (**Figure 5a**).

10) A much deeper analysis of osteopontin (OPN) is needed, and a rescue experiment using OPN in TBI IKK2-CAGFAP mice or inhibition of LCN2 should be considered to nail the mechanisms that impair wound healing following in TBI due to astrocytic NFkB activation (related to figure 9).

Response: We thank the reviewer for the comment. However, we think that such rescue experiments are beyond what is feasible for this revision, considering the long timeframe for such rescue experiments (e.g., OPN application or combination of

LCN2^{-/-} with GoF mice, obtaining the relevant licenses to adhere to animal welfare and legal regulations) .

We included some new data on Osteopontin and Lcn2 in the LoF model (**Supplementary Fig. 11**)

11) The title needs to be tempered based on existing data, and 'wound regeneration' following TBI needs to be more conclusively demonstrated with empirical evidence.

Response: We agree with the reviewer and rephrased title to "NF- κ B activation in astrocytes critically affects wound healing after traumatic brain injury".

Point-by-point response to the individual comments of the referees.

Reviewer #2 (Remarks to the Author):

I thank the authors for addressing my point-by-point concerns.

I just have one final query. The way the glial scar is quantified in new Figure 4, measuring GFAP thickness around the wound core is clearly challenging. It would not be clear where to measure the core from and which 'edge' of the 'scar' to end at. These are not nice concentric spheres and at day 3 the scar edge is almost impossible to define. The fact that this was done in a blinded manner is critical and the process would be extremely variable.

How then in Fig 4c can 'GFAP expression thickness' around the core be exactly the same measurement across three time points, with zero variability in the IKK-CA group? The lack of any variation between animals seems unlikely.

Response:

We thank the reviewer for the very valuable comment. We agree that measuring the GFAP thickness around the injury is quite challenging and, as the reviewer correctly noted, the scar edge cannot be clearly defined at day 3. For this reason, we did not quantify the GFAP scar at 3 dpi and restricted our analysis to the 7- and 15-day time points after TBI.

The reviewer's point regarding the precise location of scar quantification is well taken. To address this, we also quantified the scar by calculating the ratio of the injury size to the total area covered by intense GFAP signal, which defines the GFAP scar (the yellow dotted outline in the Figure 4). As shown in Figure 4f, this ratio was significantly lower for the IKK2-CA^{GFAP} group compared to the control group, further supporting our previous conclusion that GFAP scar formation is impaired in the IKK2-CA^{GFAP} model.

Regarding the reviewers' concern about the apparent lack of variability in GFAP expression thickness in the IKK2-CA^{GFAP} group, we apologize for the misunderstanding. Under wildtype conditions following TBI, GFAP expression is largely restricted to reactive astrocytes within the grey matter of the cerebral cortex near the injury site. In contrast, IKK2-CA^{GFAP} animals, which exhibit constitutive activated IKK2/NF- κ B signaling in astrocytes, display widespread GFAP expression that extends through the entire cortex, even under sham conditions. Therefore, we arbitrarily defined "GFAP expression thickness" as the maximal measurable extend of GFAP expression from the injury core and laterally across the ipsilateral hemisphere. Because GFAP+ astrocytes in the IKK2-CA^{GFAP} group cover the entire grey matter, this metric naturally shows no variability between individual animals. In control mice however, GFAP expression is confined to the grey matter surrounding the injury site and exhibits a well-defined boundary. Thus, the "GFAP expression thickness" in the control group varied across animals and time points. Only at 7dpi does the GFAP expression extend across the entire ipsilateral hemisphere, resulting in values comparable to the maximum (orange arrows in Figure 4).

Reviewer #3 (Remarks to the Author):

NCOMMS-24-50636A: In the revised manuscript the authors addressed several of the points raised in the prior review. New data are provided using flow cytometry of various immune population in the injured brain in the genetic models, but there are concerns about the robustness of the data based on unusual gating strategies and use of % in data presentation. The revised manuscript is also very complex and is challenging to follow due to the inclusion of so much new additional data. Simplification of the major results/findings to increase focus and access for the reader is recommended.

Below are some comments about revision 1.

1. For flow cytometry studies the live/dead staining and gating strategy used for cellular populations should be included in supplemental figures for transparency and rigor.
2. Supplemental figure 4. The annotation of the statistical comparisons in the NNS and Catwalk data are confusing (some use colour, others do not...). It would be advisable to make the data more accessible to readers by improving the statistical comparisons in this figure.
3. The placement of gates in new flow cytometry studies when comparing groups of mice is problematic (Fig 8a). The same gate needs to be used in each group to investigate phenotypic changes in immune populations due to genotype.
4. The absolute numbers for immune populations in the injured brain should be presented. Are these similar to %s shown in Figure 9c-e? It is surprising that so many peripherally derived monocytes/neutrophils are being recruited so late after TBI at 30dpi. It would be informative to see the absolute numbers across the genetic models to get a clear picture of this dynamic response.
5. The studies remain highly descriptive. The mechanistic insight about wound healing are touched upon in the final figure but left open-ended. The discussion focuses on osteopontin, but this mechanism was not examined experimentally. This is a missed opportunity to provide true mechanistic insight on role of astrocytic NfKB role in wound healing following TBI.
6. The addition of large amounts of new data in the revision means that the manuscript is extremely dense and hard to navigate for readers. This leads to a lack of focus. The authors are encouraged to simplify things and focus on critical take home messages in the results.

Response:

We thank the reviewer for the very helpful comments and for highlighting that the revised manuscript had become overly complex due to the extensive addition of new additional data. In response, we have redesigned and rewritten the entire Results section, focusing on the core messages of the study as recommended. The main findings are now presented in 8 primary figures, supplemented by 15 additional supplementary figures.

1. We agree with the reviewer that the live/dead staining and gating strategy used for cellular populations should be clearly presented. We have now consolidated this information - previously scattered across multiple figures- into a single, dedicated Supplementary Figure 11.

2. We thank the reviewer for this suggestion. We have reformatted the statistical comparisons for the NSS and Catwalk data in a uniform manner to ensure consistent and clearer presentation. Detailed descriptions of the statistical comparisons are now provided in the figure legends. Furthermore, NSS data have been moved into the main figure (now Figure 4b), while the Catwalk data are retained in Supplementary Figure 4.

3. We appreciate the reviewer's attention to this critical issue. All flow cytometry gates have been evaluated and re-adjusted across experimental groups to ensure valid comparisons of immune cell phenotypes between the different genotypes (see Figure 7).

4. We fully agree that absolute cell counts for immune populations in the injured brain are essential to get a clear picture of the dynamic post-traumatic immune response. To strengthen the manuscript, we conducted additional flow cytometry experiments and performed comprehensive cell count analyses across genotypes and time points post TBI.

These data now clearly show that the total infiltrating cell numbers are already significantly increased in IKK2-CA^{GFAP} mice even in the sham situation and continue to increase over time (until day 30) depending on the specific immune population, as seen particularly for lymphoid cells (see Figure 7j/l/n).

5. We thank the reviewer for emphasizing the importance of rescue experiments to further translate our findings in pre-clinical therapeutical approaches in future studies. We have already embarked on this extremely challenging endeavor and initiated preliminary work in this direction. However, this is beyond what is feasible for the second revision period considering the long timeframe and extensive preparatory steps required to conduct meaningful rescue experiments including securing new animal licenses and performing pilot studies to optimize application techniques and dose-response relationships (e.g., intranasal recombinant OPN application with or LCN2 neutralizing antibodies).

6. We agree with the reviewer that the previous revision was extremely dense and difficult to navigate for readers. We have therefore carefully and thoroughly reorganized the Results section: all principal findings are presented across eight main figures, while the supporting and control data, much of which directly addresses the reviewers' comments, are placed in 15 supplementary figures. We have also carefully revised the schematic summary of our findings to make the overall narrative clearer and more accessible to the reader (supplementary figure 16).